

# RAP-Net: Region Attention Predictive Network for Precipitation Nowcasting

Zheng Zhang[★1], Chuyao Luo [★1], Shanshan Feng[1], Rui Ye[1], Yunming Ye[1], and Xutao Li[1]

[1]Dept. of Computer Science, Harbin Institute Technology, Shenzhen, China
★These authors have contributed equally to this work.

**Correspondence:** lixutao@hit.edu.cn (Xutao Li)

**Abstract.** Natural disasters caused by heavy rainfall often cause huge loss of life and property. Hence, the task of precipitation nowcasting is of great importance. To solve this problem, several deep learning methods have been proposed to forecast future radar echo images and then the predicted maps are converted to the distribution of rainfall. The prevailing spatiotemporal sequence prediction methods apply ConvRNN structure which combines the Convolution and Recurrent neural network. Although ConvRNN methods achieve remarkable success, they ignore capturing both local and global spatial features simultaneously, which degrades the nowcasting in regions of heavy rainfall. To address this issue, we propose a Region Attention Block (RAB) and embed it into ConvRNN to enhance forecasting in the area with strong rainfall. Besides, the ConvRNN models are hard to memorize longer historical representations with limited parameters. To this end, we propose Recall Attention Mechanism (RAM) to improve the prediction. By preserving longer temporal information, RAM contributes to the forecasting, especially in the middle rainfall intensity. The experiments show that the proposed model Region Attention Predictive Network (RAP-Net) significantly outperforms state-of-the-art methods.

## 1 Introduction

Precipitation nowcasting has vital influence in the field of transportation, agriculture, tourism industry, and city alarming. Due to the higher spatial and temporal resolution of the radar echo image, it is effective for forecasting the distribution of rainfall by predicting the future radar echo maps and converting each pixel to the rainfall intensity according to the Z-R relationship Shi et al. (2017). Therefore, precipitation nowcasting is often defined as a spatiotemporal prediction problem.

Traditional approaches of precipitation nowcasting are motion field-based methods. The specific process can be briefly divided into three parts. First, the motion field is estimated by variational radar echo tracking methods such as optical flow Woo and Wong (2017). Second, the future radar reflectivities are advected by a semi-Lagrangian advection scheme under the assumption of stationary movement. Third, the performance of forecasts is evaluated by comparing to ground truth. However, these methods do not exploit abundant historical observations.

To overcome the limitation, some deep learning-based methods have been proposed to handle precipitation nowcasting Shi et al. (2017); Ayzel et al. (2020); Li et al. (2021). They usually build a mapping from previous observations to future maps by learning from the abundant historical radar data. Generally, the prevailing approaches utilize the structure of ConvRNN,





which combines the Convolution Neural Network (CNN) and Recurrent Neural Network (RNN). Furthermore, to enhance the spatiotemporal representation ability, other types of neural networks such as Spatial Transformer Network (STN) Shi et al. (2017), Deformable Convolution Network (DCN) Wu et al. (2021) and Attention Module Lin et al. (2020) are introduced in the ConvRNN unit and obtain better performance.

However, existed ConvRNN models confront three drawbacks: 1) The convolution employed in the current input only extracts the local features instead of the large-scale representation due to fixed kernel size. It may lead to that useful information beyond the visual field of convolution cannot be captured and thus degrades the performance. 2) The convolution applied in the previous hidden states only transmits local previous representation to the current states, which causes historical spatial information cannot be fully used. 3) The update process of temporal memory limits the long-term spatiotemporal representation preservation. Thus, the information including high echo reflectivity is easily dropped. Although some remedial solutions Wang et al. (2018b); Luo et al. (2021) based on attention mechanism are proposed, they are hard to be applied in large-scale inputs and long-term predictions due to the limitation of space occupation.

To address the first two problems, we propose a Region Attention Block (RAB) and embed it into the input and hidden state, respectively. It simultaneously exploits the global spatial representation and preserves the local feature. RAB classifies each feature map into equal-sized tensors, where the similar semantics gathered in the same tensor. Then, the attention module is used to interact with the contents of all semantics. To this end, the large-scale feature map can be captured from the global view, and meanwhile maintain local representations. Therefore, the large-scale spatial feature of the current input and previous hidden states can be preserved. Moreover, to capture the long-term spatiotemporal dependency of representation without increasing parameters, we present the Recall Attention Mechanism (RAM) to retrieval all historical inputs. More rainfall information is captured by this component. By combining these modules, the performance for heavy and middle rainfall can be significantly improved. In brief, the main contributions of the paper are summarized as follows:

1. We first propose a new attention method, namely Region Attention Block (RAB), to capture both global and local spatial features simultaneously to improve the spatial expressivity of feature maps.

2. We embed the RAB into current inputs and previous hidden states to obtain the large-scale spatial information from the global view and persevere different semantics at the same time. For the same echo with large-scale size and long-range movement between the adjacent time, more useful spatial information can be extracted, which leads to more accurate predictions in those regions with heavy rainfall.

3. We propose the Recall Attention Mechanism (RAM) to retrieval all historical inputs with limited parameters. The representation of middle and strong rainfall intensity can be preserved in the predicted unit.

## 2   Related Work

Traditional methods Pulkkinen et al. (2019) mainly focus on estimating the motion field between the adjacent radar maps and then the next prediction can be extrapolated based on this movement. Here, the motion field describes the direction and distance





of each pixel that need to be moved at the next moment. To obtain the movement, Tracking Radar Echoes by Correlation (TREC) Wang et al. (2013) divides the whole radar maps into serval equal-sized boxes and calculates the motion vector of each pair's box center by searching the highest correlation between boxes at the adjacent time steps. Another type of approach is the

optical flow-based method Woo and Wong (2017). It calculates the motion field under pixel level based on the assumption that the brightness of pixels remains unchanged. Upon the idea, many algorithms Ryu et al. (2020) are developed to apply the radar maps with the large movement vector. However, the invariant brightness assumption conflicts with the realistic movements of hydrometeors and massive historical data are utilized.

To overcome it, many deep learning-based methods Wang et al. (2017, 2019); Trebing et al. (2021) are proposed to predict

the radar sequence without the above unreasonable assumption. Most of methods commonly exploit the structure of ConvRNN. It combines Convolution Neural Network (CNN) and Recurrent Neural Network (RNN) to preserve the spatiotemporal feature of the historical sequence. Furthermore, Wang et.al added a spatial memory in predicted unit Wang et al. (2017, 2018a) and attention mechanism in temporal memory Wang et al. (2018b) to enhance the spatiotemporal representation ability of short-term and long-term, respectively. Although these methods have remarkable performance, the visualization of their predictions

is usually burry due to the loss function and the architecture of the model Shouno (2020). To handle the issue, Generative Adversarial Network (GAN) Tian et al. (2019); Xie et al. (2020); Zheng et al. (2021) has been introduced in the ConvRNN model to improve predictive clarity. Nevertheless, the non-convergence and collapse problem would cause a negative influence on prediction. Our proposed method is different from existed deep learning methods in two aspects. In the short term, the proposed RAB can simultaneously exploit local and global spatial-temporal representations. In the long term, the RAM can

effectively recall all historical observations with limited space occupancy.

## 3  Proposed Method

### 3.1  Problem Definition

The precipitation nowcasting task can be defined as the spatiotemporal sequence prediction problem Shi et al. (2017). Based on historical observations $X_{0:t}$, it aims to forecast the future radar echo images $\bar{X}_{t+1:T}$ that have maximum probability with

ground truth $X_{t+1:T}$ as following:

$$\bar{X}_{t+1:T} = \arg\max P(X_{t+1:T}|X_0, X_1, \cdots, X_t). \tag{1}$$

In this paper, $t$ and $T$ are set to 5 and 15 respectively, which means that ten continuous radar maps need to be predicted according to five historical images.

### 3.2  Overall Architecture

The overall architecture of the proposed model RAP-Net is presented in Figure 1. It utilizes the structure of PredRNN Wang et al. (2017) and stacks several RAP-Units to generate the predictions from timestamp 2 to $T$. The red and blue arrows denote

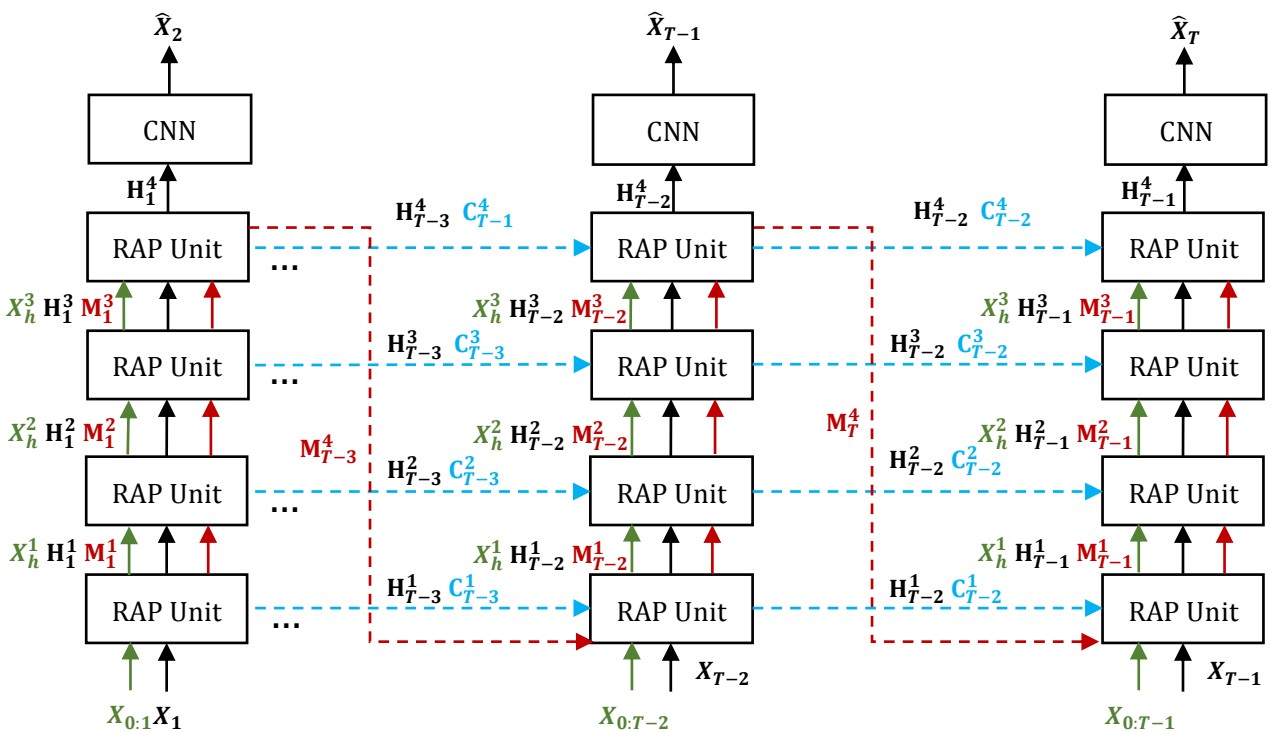

**Figure 1.** The overall architecture of the Region Attention Predictive Network (RAP-Net)

the delivering direction of spatial memory $M$ and temporal memory $C$, respectively. These two memories preserve spatial and temporal representations, respectively. Different from the PredRNN, RAP-Net exploit dissimilar data flow to transmit long-term spatiotemporal information $X_h^l$ which preserves all historical representations. Besides, we notice that the majority of ConvRNN models Wang et al. (2017, 2018a, b, 2019); Lin et al. (2020) employ similar architecture. Hence, the difference lies in their units instead of the employed architecture. In the experiment section, we will discuss and analyze the performance of different units utilized by the existed methods.

The interal structure of Region Attention Unit (RAP-Unit) is shown in Figure 2. The inputs include the current input $X_t^l$, previous hidden state $H_{t-1}^l$, temporal memory $C_{t-1}^l$, spatial memory $M_t^{l-1}$ and long-term historical representation $X_h^{l-1}$. According to Figure 1, RAP-Net is consisted of four stacked RAP-Units. At the bottom layer, $X_h^{l-1}$ represents all historical inputs $X_{0:t}$. While at other layers, $X_h^{l-1}$ is the output of previous layer. The outputs of RAP-Unit are the current hidden state $H_t^l$, spatial memory $M_t^l$, temporal memory $C_t^l$ and new long-term representation $X_h^l$. The details of calculation are presented according to following formulas:





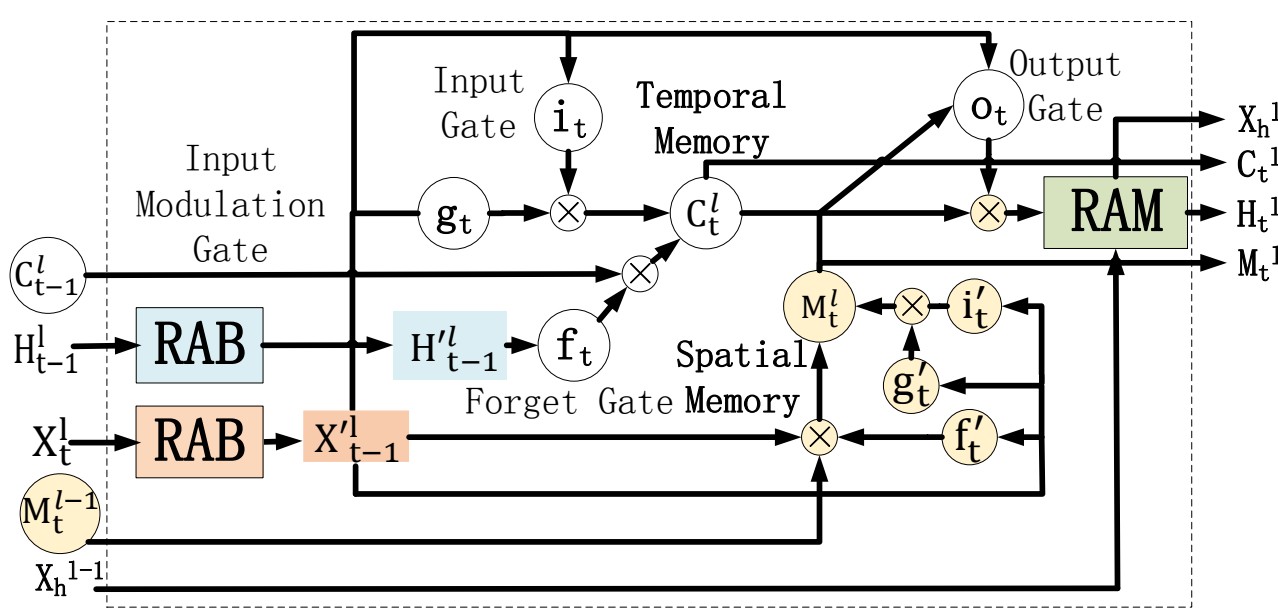

**Figure 2.** The internal structure of the Region Attention Predictive Unit (RAP-Unit)

$$X_t = RAB(X_t),$$

$$H_{t-1} = RAB(H_{t-1}),$$

$$i_t = \sigma(W_{xi} * X_t + W_{hi} * H^l_{t-1} + b_i),$$

$$g_t = tanh(W_{xg} * X_t + W_{hg} * H^l_{t-1} + b_g),$$

$$f_t = \sigma(W_{xf} * X_t + W_{hf} * H^l_{t-1} + b_f),$$

$$i'_t = \sigma(W'_{xi} * X_t + W_{mi} * M^{l-1}_t + b'_i),$$

$$g'_t = tanh(W'_{xg} * X_t + W_{mg} * M^{l-1}_t + b'_g),$$

$$f'_t = \sigma(W'_{xf} * X_t + W_{mf} * M^{l-1}_t + b'_f),$$

$$C^l_t = i_t \circ g_t + f_t \circ C^l_{t-1},$$

$$M^l_t = i'_t \circ g'_t + f'_t \circ M^{l-1}_t,$$

$$o_t = \sigma(W_{xo} * X_t + W_{ho} * H^l_{t-1} + W_{co} * C^l_t + W_{mo} * M^l_t + b_o),$$

$$H^l_t = o_t \circ tanh(W_{1\times1} * [X^l_t, M^k_t]),$$

$$H^l_t, X^1_h = RAM(H^l_t, X^{l-1}_h * W_l), \tag{2}$$



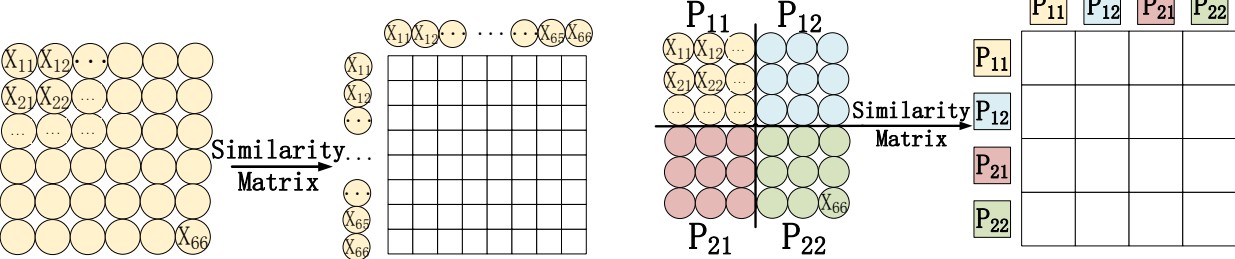

(a) The similarity matrix in terms of tradictional attention mechanism Vaswani et al. (2017).

(b) The similarity matrix in terms of the Vision Transformer Dosovitskiy et al. (2020).

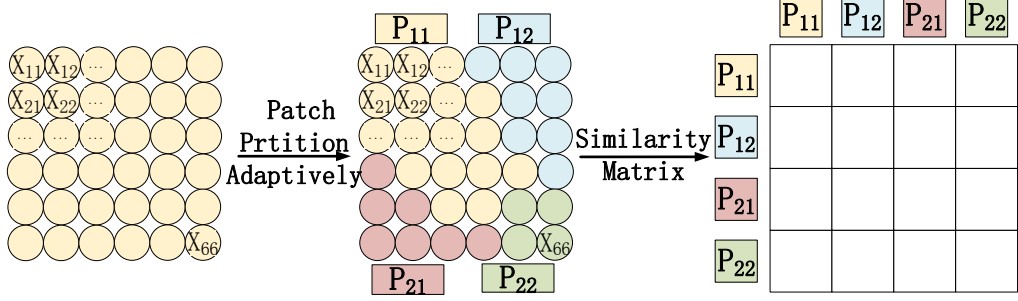

(c) The similarity matrix in terms of the Region Attention (ours).

**Figure 3.** The calculation process of similarity matrix based on three different attention methods

where '$*$' and '$\circ$' denote the convolution and Hadamard product respectively. '$i_t$','$g_t$','$f_t$','$i_t'$','$g_t'$','$f_t'$' indicate various gates, which can be viewed as intermediate variables. Here, RAB and RAM are the Region Attention Block and Recall Attention

Mechanism, respectively.

### 3.3 Region Attention Block

To address the issue, we expect that these patches can be divided adaptively and those elements with similar semantic relationships are classified into the same patch shown in Figure 3(c). To realize this idea, we propose the Region Attention Block (RAB) whose structure is illustrated in Figure 4. First, a convolution and softmax layer are employed in the input feature map

$F^i \in R^{B \times C \times H \times W}$ to generate $F^c \in R^{B \times N \times H \times W}$ for distinguishing $N$ classifications. Second, the original input $F^i$ is split $N$ groups of feature maps $F^n \in R^{N \times B \times C \times H \times W}$ by the Split module following this formula:

$$\begin{aligned} \text{Split}(F^i, F^c) = &\text{Concatenate}(\{F_{j,k,h,w}^i \cdot F_{j,n,h,w}^c | 1 < n < N, n \in Z\}, \\ &axis = 0), \end{aligned} \quad (3)$$



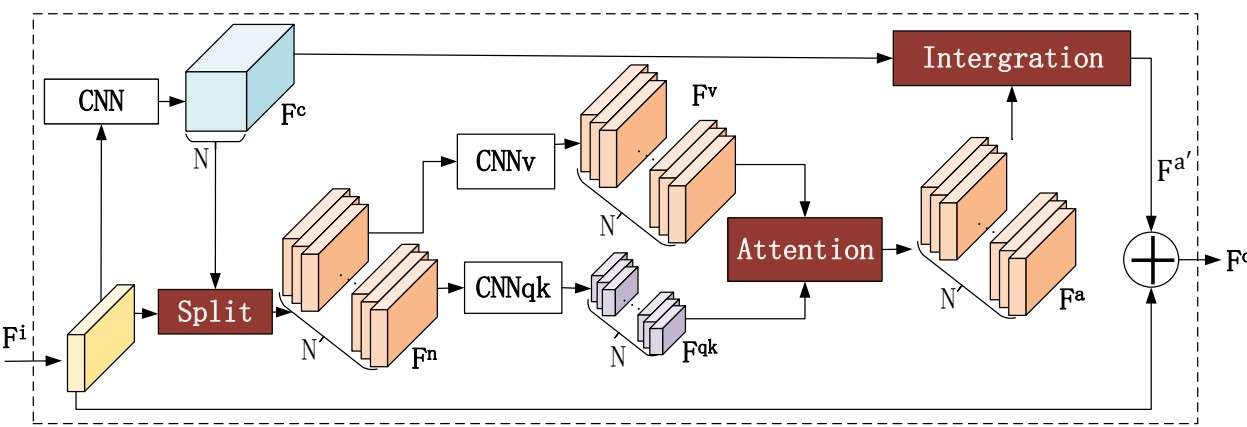

**Figure 4.** The structure of Region Attention Block (RAB).

These groups denote various semantic information extracted from different positions. Third, $F^{qk} \in R^{B \times N \times c \times h \times w}$ is convolved by $F^n$ to further exploit the feature of $F^n$ and reduce the parameters, where the $c$, $h$ and $w$ are smaller than $C$, $H$ and $W$. Besides, $F^v \in R^{N \times B \times C \times H \times W}$ are outputed by a convolution layer applied in $F^n$ to preserve original information. Forth, three different convolutions are used to generate query $Q_s$, key $K_s$ and value $V_s$ based on $F^{qk}$ and $F^v$. After flattening, $Q_s \in R^{B \times N \times c*h*w}$, $K_s \in R^{B \times N \times c*h*w}$ and $V_s \in R^{B \times N \times C*H*W}$ are fed into the spatial attention function to obtain $F^a \in R^{B \times N \times C \times H \times W}$ that have been interacted the local representation from different regions. The output after attention function Fu et al. (2019) is:

$$Attention(Q_s, K_s, V_s) = softmax(\frac{f(Q_s, K_s^T)}{\sqrt{d_k}})V_s, \tag{4}$$

where $f$ denotes dot-product and $d_k$ is the dimension of key $K_s$. $f(Q_s, K_s^T) \in R^{B \times N \times N}$ is the similarly matrix of various semantics in different regions. Fifth, an Integration module is utilized to integrate $F^a$ based on the $F^c$ and get the result $F^{a'}$ by this equation:

$$F^{a'} = Integration(F^a, F^c)$$
$$= \sum_{n=1}^{N} F^a_{j,k,h,w} \cdot F^c_{j,n,k,h,w}. \tag{5}$$

where, $F^{a'}$ has the same size as input feature map $F^i$. Finally, the structure of ResNet He et al. (2016) is introduced to deeply exploit spatial feature and achieve the final result $F^o \in R^{B \times N \times C \times H \times W}$. In summary, the calculation process is described by the following formulas:





$$F^c = softmax(F^i * W_c),$$

$$F^n = \text{Split}(F^i, F^c),$$

$$F^{qk} = F^n * W'_{qk},$$

$$F^v = F^n * W'_v,$$

$$F^a = \text{Attention}(F^{qk} * W_q, F^{qk} * W_k, F^v * W_v),$$

$$F^{a'} = \text{Integration}(F^a, F^c),$$

$$F^o = F^i + F^{a'}. \tag{6}$$

### 3.4 Recall Attention Mechanism

To capture the temporal long-dependencies of representation, Wang et.al Wang et al. (2018b) embedded the spatial attention module in the updating of temporal memory. However, it has two limitations: 1) It saves abundant history temporal memories, which leads that the number of parameters easily exceeds the space occupancy as lead time goes. 2) The temporal memory has lost some information during the generation of various gates. Therefore, the preserved previous representation fails to capture all the information and long-term spatiotemporal expressivity is limited.

To address these issues, we propose the Recall Attention Mechanism (RAM) to enhance the long-term spatiotemporal representation ability with fixed space occupation as Figure 5 shows. First, we build an empty long-memory feature map $X_h^0 \in R^{B \times T \times C \times H \times W}$ in the bottom layer and feed into the current input $X_t$ continually. Note that the $X_h^0$ contains all original previous inputs $X_{0:t} \in R^{B \times T \times C \times H \times W}$. Second, a convolution neural network is employed to extract the feature of $X_h^0$ and output the long-memory hidden state $X_h^1 \in R^{B \times T \times C \times H \times W}$. Lastly, $X_h^1$ and the output $H_t'^l$ of RAP-Cell feed into the channel attention module to generate new hidden states, where the $X_h^1$ can be regarded as the key $K_c \in R^{B \times T*C \times H*W}$ and value $V_c \in R^{B \times T*C \times H*W}$, and the $H_t'$ represents the query $Q_c \in R^{B \times C \times H*W}$. The formula of channel attention is shown as:

$$\text{Attention}(Q_c, K_c, V_c) = softmax(\frac{f(Q_c, K_c^T)}{\sqrt{d_k}})V_c, \tag{7}$$

where, the $f$ denotes dot-product and $d_k$ is the dimension of key $K_c$. $f(Q_c, K_c^T) \in R^{B \times T \times T*C}$ is the similarly matrix between channels of $Q_s$ and channels of $K_c$. In this mechanism, the new output $H_t$ has recalled all original historical representation and long-term dependencies can be effectively preserved. Besides, the size of the long-memory feature map $X_h$ is fixed at any time step. In the next layer, the input of the long-memory hidden state is the $X_h^1$ from the bottom layer. To this end, the long-term historical representation can be delivered to the next layer.



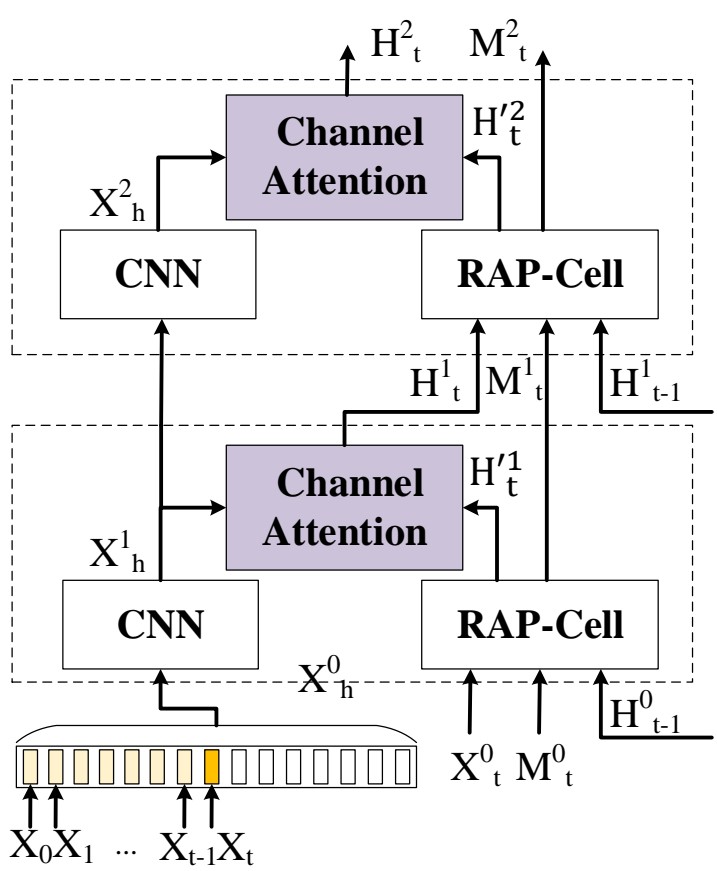

**Figure 5.** The structure of Recall Attention Mechanism (RAM).

## 4 Experiments

### 4.1 Dataset

The dataset is collected from the CIKM AnalytiCup2017 competition, which covers the whole area of Shenzhen city, China. For convenience, we name this public dataset to RadarCIKM. RadarCIKM has a training set and test set with 10,000 and 4,000 sequences, respectively. There are 2,000 sequences randomly sampled from the training set to build the validation set. Each sequence contains 15 continual observations within 90 minutes, where the spatial and temporal resolution of each map is 1km × 1km and six minutes, respectively. The range of each pixel is from 0 to 255 and each pixel $p$ can be converted to radar reflectivity (dBZ) by Z-R relationship as follows:





$$dBZ = p \times \frac{95}{255} - 10. \tag{8}$$

## 4.2 Evaluation Metrics

In this paper, in addition to common measurements such as Structural Similarity (SSIM) and Mean Absolute Error (MAE) in video prediction, we also utilize the Heidke Skill Score (HSS) and Critical Success Index (CSI) that are commonly used in precipitation nowcasting task. The HSS evaluates the fraction of correct forecasts after eliminating random predictions. The CSI measures the number of correct forecasts divided by the total number of occasions when the rainfall events were forecasted or observed. Specifically, the prediction and ground truth are converted to binary matric based on a threshold $\tau$. When the value of dBZ is larger than $\tau$, it is set to 1 otherwise to 0. Next, the number of the True-Positive (TP, prediction=1 and truth = 1), False-Negative (FN, prediction = 0 and truth=1), False-Positive (FP, prediction=1 and truth=0) and True-Negative (TN, prediction=0 and truth=0) are counted. Finally, the HSS and CSI can be calculated by the following formulas:

$$HSS = \frac{2(TP \times TN - FN \times FP)}{(TP + FN)(FN + TN) + (TP + FP)(FP + TN)}, \tag{9}$$

$$CSI = \frac{TP}{TP + FN + FP}. \tag{10}$$

### 4.3 Parameters Setting

The proposed RAP-Net takes five previous radar echo maps as inputs and outputs ten predictions. It utilizes four layers RAP-Units as shown in Figure 1, where the number of patches is set to 64. The Adam optimizer is applied to train our model with learning rate 0.0004. Besides, the early stopping and scheduled sampling strategies are also used to optimize our model. The loss function combines the L1 and L2 to train SST-LSTM. All experiments are implemented in Pytorch and conducted on NVIDIA 3090 GPUs.

### 4.4 Result and Analysis

Tables 1 and 2 show the results of all models. The best results are in boldface and the second best scores are underlined. We find that the RAP-Net achieves the smallest error and the highest structural similarity according to the MAE and SSIM. It is observed that that our model outperforms other models in terms of the comprehensive performance. Besides, the proposed model has significant superiority especially for the nowcasting in heavy rainfall regions. Because the HSS and CSI keep the top position in the middle and high thresholds (20 dBZ and 40 dBZ). For the state-of-art method, PFST-LSTM Luo et al. (2020), all measurements of it are exceeded by RAP-Net, which shows the performance of our model furthermore. Comparing with PredRNN, PredRNN++, and RAP-Net, we can see that they have similar SSIM due to applying the same architecture. However, the other evaluation scores of RAP-Net are significantly higher than PredRNN and PredRNN++, which implies the benefit of



**Table 1.** Comparison results on RadarCIKM in terms of HSS and MAE

| Methods | HSS ↑ | | | | MAE ↓ |
|---|---|---|---|---|---|
| | 5dBZ | 20dBZ | 40dBZ | avg | |
| ConvLSTM Xingjian et al. (2015) | 0.7031 | 0.4857 | 0.1470 | 0.4453 | 5.97 |
| ConvGRU Shi et al. (2017) | 0.6816 | 0.4827 | 0.1225 | 0.4289 | 6.00 |
| TrajGRU Shi et al. (2017) | 0.6809 | 0.4945 | 0.1907 | 0.4553 | 5.90 |
| DFN Jia et al. (2016) | 0.6772 | 0.4719 | 0.1306 | 0.4266 | 6.03 |
| PredRNN Wang et al. (2017) | 0.7082 | 0.4915 | 0.1639 | 0.4606 | 5.42 |
| PredRNN++ Wang et al. (2018a) | 0.7061 | 0.5047 | 0.1710 | 0.4548 | 5.44 |
| MIM Wang et al. (2019) | 0.7052 | 0.5166 | 0.1858 | 0.4692 | 5.47 |
| PhyDNet Guen and Thome (2020) | 0.6741 | 0.4709 | 0.1832 | 0.4427 | 6.25 |
| SA-ConvLSTM Lin et al. (2020) | **0.7118** | 0.4861 | 0.1582 | 0.4520 | 5.71 |
| PFST-LSTM Luo et al. (2020) | 0.7045 | 0.5071 | 0.2218 | 0.4778 | 5.82 |
| CMS-LSTM Chai et al. (2021) | 0.6835 | 0.4605 | 0.1720 | 0.4387 | 5.95 |
| RAP-Net | 0.7117 | **0.5116** | **0.2293** | **0.4842** | **5.37** |

**Table 2.** Comparison results on RadarCIKM in terms of CSI and SSIM

| Methods | CSI ↑ | | | | SSIM ↑ |
|---|---|---|---|---|---|
| | 5dBZ | 20dBZ | 40dBZ | avg | |
| ConvLSTM Xingjian et al. (2015) | 0.7663 | 0.4092 | 0.0801 | 0.4186 | 0.6334 |
| ConvGRU Shi et al. (2017) | 0.7522 | 0.3952 | 0.0657 | 0.4043 | 0.6338 |
| TrajGRU Shi et al. (2017) | 0.7466 | 0.4028 | 0.1061 | 0.4185 | 0.6424 |
| DFN Jia et al. (2016) | 0.7489 | 0.3771 | 0.0704 | 0.3988 | 0.6268 |
| PredRNN Wang et al. (2017) | 0.7692 | 0.4051 | 0.0901 | 0.4215 | 0.6887 |
| PredRNN++ Wang et al. (2018a) | 0.7642 | 0.4176 | 0.0940 | 0.4253 | 0.6851 |
| MIM Wang et al. (2019) | 0.7628 | 0.4279 | 0.1034 | 0.4313 | 0.6796 |
| PhyDNet Guen and Thome (2020) | 0.7402 | 0.4003 | 0.1017 | 0.4141 | 0.6443 |
| SA-ConvLSTM Lin et al. (2020) | **0.7725** | 0.4161 | 0.0870 | 0.4252 | 0.6709 |
| PFST-LSTM Luo et al. (2020) | 0.7680 | 0.4175 | 0.1257 | 0.4371 | 0.6367 |
| CMS-LSTM Chai et al. (2021) | 0.7567 | 0.3788 | 0.0948 | 0.4101 | 0.6496 |
| RAP-Net | 0.7666 | **0.4305** | **0.1307** | **0.4426** | **0.7019** |

RAP-Unit. Lastly, we notice the SA-ConvLSTM Lin et al. (2020) gets the best HSS and CSI in the lowest threshold (5 dBZ). Nevertheless its performance is poor in the highest threshold (40dBZ), which implies the Region Attention can improve the prediction in the area with high radar echo compared to traditional attention mechanism.







(a) HSS $\tau=5$;

(b) CSI $\tau=5$;

(c) HSS $\tau=20$;

(d) CSI $\tau=20$;

(e) HSS $\tau=40$;

(f) CSI $\tau=40$;

**Figure 6.** The HSS and CSI scores of different nowcase lead time values. (Best view in color)





**Table 3.** Ablation results on RadarCIKM in terms of HSS and MAE

| Methods | HSS ↑ | | | | MAE ↓ |
|---------|-------|-------|-------|------|-------|
| | 5dBZ | 20dBZ | 40dBZ | avg | |
| PredRNN | 0.7082 | 0.4915 | 0.1639 | 0.4545 | 5.42 |
| RAP-Cell$_x$ | 0.7102 | 0.5042 | 0.1754 | 0.4633 | 5.36 |
| RAP-Cell$_h$ | 0.7149 | 0.4967 | 0.1753 | 0.4623 | **5.32** |
| RAP-Cell | **0.7234** | 0.4757 | 0.2283 | 0.4758 | 5.64 |
| RAP-Net | 0.7117 | **0.5116** | **0.2293** | **0.4842** | 5.37 |

To show the performances of various models at different nowcasting lead times, Figure 6 presents the HSS and CSI curves
w.r.t different lead times under all thresholds. We observe that both HSS and CSI scores of all models become decrease
as the lead time increases, which shows the difficulty of long-term predictions. Among these models, RAP-Net achieves
notable superiority in the middle and late stages of the nowcasting period at the highest threshold. Especially in the last
prediction, all baseline methods trend to get the same worse result. The RAP-Net remarkably outperforms other models. It
implies that the introduction of RAB and RAM in the proposed model contributes to generating long-term predictions with
heavy rainfall region. Although the performance of RAP-Net would be degraded when the threshold becomes small, it still has
competitiveness compared to other models.

Figure 7 shows an example of predictions from these models. The various colors denote the different ranges of reflectivity
according to the color bar in the bottom of Figure 7. From the ground truth in the first row, the rainfall event is obviously the
trend of increasing the rainfall intensity. However, only our model can forecast this trend and keep the intensity of the regions.
The RAP-Net can generate a high reflectivity area, which can also explain why our model can achieve the highest evaluate
index HSS and CSI in the middle and high thresholds.

### 4.5 Ablation Study

To investigate the influence of various modules, we conduct an ablation study to discuss the effectiveness of Region Attention
Block to the current input and the last hidden state. The result of evaluations is shown in Tables 3 and 4. RAP-Cell$_x$ and
RAP-Cell$_h$ denote the PredRNN model embedding the RAB into the input and hidden state, respectively. RAP-Cell model is
the combination of RAP-Cell$_x$ and RAP-Cell$_h$, and can also be regarded as RAP-Net without RAM. The results of RAP-Cell$_x$
and RAP-Cell$_h$ are higher than PredRNN, which shows the advantage of introducing Region Attention Block. Specially, the
RAP-Cell$_h$ significantly reduces the error according to MAE. Besides, the HSS, CSI and SSIM of RAP-Cell have significant
improvements particularly when threshold $\tau$ is 40dBZ, which implies that RAB simultaneously employed in the input and
hidden state contributes to the prediction in the heavy rainfall regions. Moreover, by comparing the RAP-Cell and RAP-Net,
we find that the RAM can enhance the accuracy of nowcasting especially in the area with middle-intensity rainfall.







**Figure 7.** The first row is the ground truth and reminders are the predictions of various methods on an example from the RadarCIKM dataset (Best view in color)





(a) HSS $\tau$=5;

(b) CSI $\tau$=5;

(c) HSS $\tau$=20;

(d) CSI $\tau$=20;

(e) HSS $\tau$=40;

(f) CSI $\tau$=40;

**Figure 8.** The performance changes against different nowcast lead time in terms of HSS and CSI scores in ablation study. (Best view in color)





**Table 4.** Ablation results on RadarCIKM in terms of CSI and SSIM

| Methods | CSI ↑ | | | | SSIM ↑ |
|---|---|---|---|---|---|
| | 5dBZ | 20dBZ | 40dBZ | avg | |
| PredRNN | 0.7692 | 0.4051 | 0.0901 | 0.4215 | 0.6887 |
| RAP-Cell$_x$ | 0.7747 | 0.4235 | 0.0967 | 0.4316 | 0.6965 |
| RAP-Cell$_h$ | 0.7772 | 0.4138 | 0.0967 | 0.4292 | 0.7009 |
| RAP-Cell | **0.7817** | 0.4143 | 0.1300 | 0.4420 | **0.7036** |
| RAP-Net | 0.7666 | **0.4305** | **0.1307** | **0.4429** | 0.7019 |

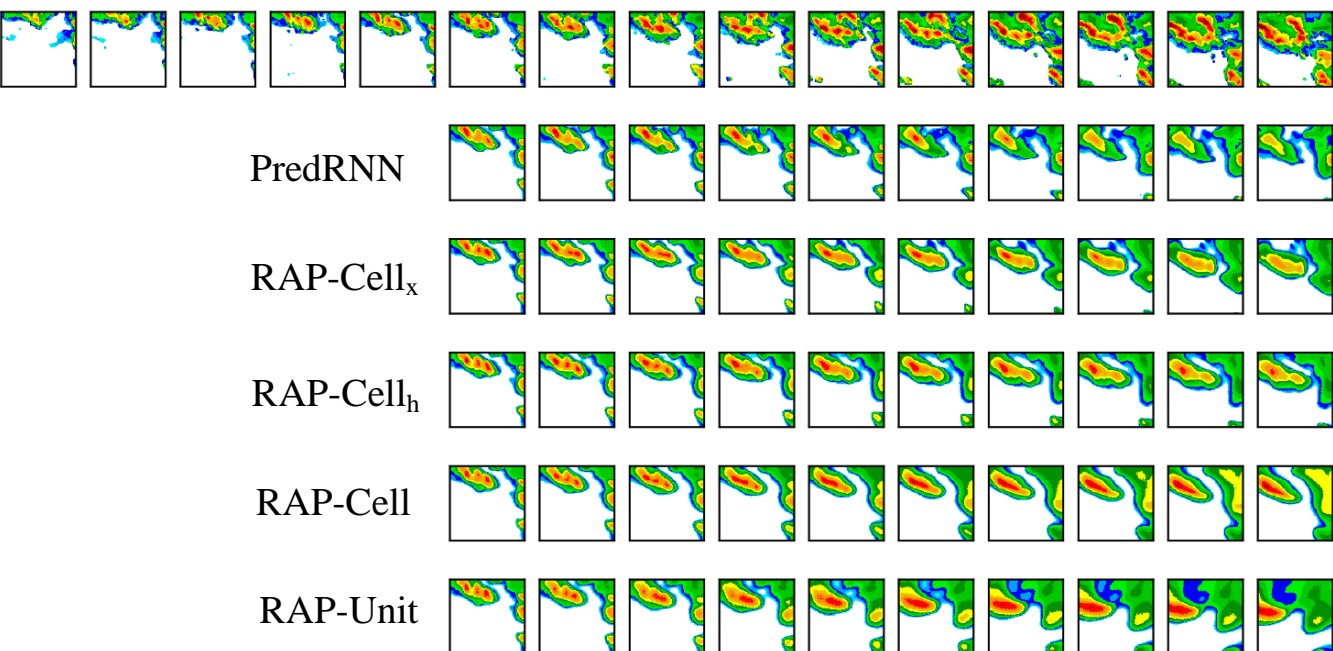

**Figure 9.** The first row is the ground truth and reminders are the predictions of different methods on an example from the RadarCIKM dataset (Best view in color)

Similarly, we also plot Figure 8 to show experimental results of all models against different nowcast lead times. We can see that RAP-Net delivers more promising result when the threshold increases, which demonstrates the effectiveness of combining RAB and RAM in terms of long-term prediction in high reflectivity area. The performance of RAM can be shown by com-
230 paring RAP-Cell and RAP-Net. We notice that the introduction of RAM can improve the prediction in the region of middle rainfall intensity. Besides, RAP-Cell$_x$ and RAP-Cell$_h$ embed RAM in the current input and the hidden state, respectively. Their performance is better than the original model PredRNN, especially in 20dBZ threshold. It shows the superiority of RAM.

We also show predictions of different methods for a given sample in Figure 9. We find that RAP-Cell can generate the red area which is reflected by better evaluation indexes of HSS and CSI in the highest threshold. However, all forecasts except for

RAP-Net have a gap in the radar echo block, which is obviously different from the ground truth. The improvement of prediction
in middle rainfall intensity can be owed to the embedding RAM.

## 5 Conclusions

In this paper, we propose the RAP-Net to handle the precipitation nowcasting task. On the one hand, it embeds the Region
Attention Block to enhance the local and global spatial representation ability simultaneously by extracting and delivering the
240 features in ConvRNN. The improvement can significantly enhance the accuracy especially in those regions with heavy rainfall.
On the other hand, we introduce the Recall Attention Mechanism to improve the temporal expressivity in the long term. It
can preserve and retrieve longer historical information and effectively enhance the performance of prediction, particularly
for the middle rainfall intensity. We conduct extensive experiments to evaluate the performances of most ConvRNN models.
Empirically, RAP-Net can preserve regions of heavy intensity in long-term predictions. It shows the effectiveness of RAB and
245 RAM to improve forecasting. The ablation study independently measures the influence of these two modules. The RAB is able
to enhance the accuracy in the high threshold and RAM can improve the prediction in the middle threshold.

Currently, most of existed methods focus on radar echo maps prediction based on a single altitude layer. The variety and
movement of echo not only need to consider the previous sequence in the same layers but also need to use different altitude
layers. For future work, we will consider integrating other layers' historical information to improve the forecasting and we will
perform further experiments on multi-channel RAP-Net based on multi-layers radar echo images.

*Code availability.* The source code and pretrained model of RAP-Net are available at: https://doi.org/10.5281/zenodo.5979275

*Data availability.* The data is available at: https://doi.org/10.5281/zenodo.5979275

*Author contributions.* Conceptualization, methodology, and writing investigation was performed by Zheng Zhang and Chuyao Luo. Rui Ye
contributed to the project administration, and visualization. Shanshan Feng and Xutao Li contributed to writing, review, editing, data curation,
and validation. Yunming Ye contributed to supervision, visualization, investigation, resources and funding acquisition.

*Competing interests.* The authors declare that they have no conflict of interest

*Acknowledgements.* This work was supported in part by the Shenzhen Science and Technology Program, China under Grant JCYJ20200109113014456
and JCYJ20210324120208022





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
