# Peer review of "RAP-Net: Region Attention Predictive Network for Precipitation Nowcasting"

_Geoscientific Model Development, 2022_

## Referee Comment (RC1)

**Review**

The authors improve the model based on PredRNN and design a Region Attention Predictive Network. Through experiments, it can be confirmed that the model has some improvements in middle intensity rainfall prediction. However, there are still some problems need to be solved:

(1) The mechanism of RAB and RAM improving the prediction accuracy of middle intensity rainfall rather than other rainfall is not explained.

(2) The $\hat{X}_2$, $\hat{X}_{T-1}$, $\hat{X}_T$ in Figure 1 is not explained.

(3) The authors mention the improvement of RAM parameter performance, but lack discussion of RAB performance. In RAB, the global information is strengthened by self-attention mechanism, which brings more computation and parameters, and improves the prediction accuracy. Compared with the increase of computation, does this increase in accuracy meet expectation?

(4) In Figure 3, what are the specific advantages of regional attention similarity matrix over other attention?

(5) Page 7 Line 128, "…$Q_s \in R^{B \times N \times c*h*w}$, $K_s \in R^{B \times N \times c*h*w}$ and $V_s \in R^{B \times N \times C*H*W}$ are fed…". What does tensor range mean?

(6) Please give the specific parameters of RAB and RAM.

(7) Page 8 Line 159. The authors regard $X_h^1$ as $K_c$ and $V_c$, then regard $H_t'$ as $Q_c$. Please explain the reason for this.

(8) Page 8 Line 163. "…the new output $H_t$ has recalled all original historical representation and long-term dependencies can be effectively preserved. Besides, the size of the long-memory feature map $X_h$ is fixed at any time." The reason why self-attention can achieve this effect is not explained.

(9) As for RAM having more advantages than the recall mechanism of EIDETIC 3D LSTM, the paper lacks important experiment to prove it.

(10) What is the resolution of the image processed before experiments?

Minor comments:

(1) Page 3 Line 85, "It utilizes the structure of PredRNN …". When the abstract and previous part have been around ConvRNN method, PredRNN is mentioned here.

Is it possible to explain the relationship between them?

(2) Page 11 Line 202, "… which implies the Region Attention can improve …". Is it possible to write RAB and RAM together, not just Region Attention?

---

## Referee Comment (RC3)

Review

To predict weather radar image sequences, the authors propose the model RAP-Net that add attention modules (RAB and RAM) in a ConvRNN model, in order to improve forecasting in the area with heavy rainfall (RAB) and improve the long-term spatiotemporal representation ability (RAM).

1. In general, rainfall intensity is classified as light, moderate and heavy (AMS glossary "Rain": https://glossary.ametsoc.org/wiki/Rain), not as "strong" (line 7) or "middle" (lines 10, 44, 53 and 243). Please fix this throughout the text.
2. In line 22, what are the limitations of the traditional approach you are referring to?
3. In line 39, the sentence "where the similar semantics gathered in the same tensor." sounds incomplete.
4. I suggest reviewing all equations to simplify the notation and unify the letters of all equations (sections 3.2, 3.3 and 3.4).
   a. The notation adopted by the terms of equations is confusing (sections 3.2, 3.3 and 3.4):
      1. In line 89, what are the letters "$l$" and "$h$" in $X_h^l$?
      2. Is $X_h^l = X_t$?
      3. Etc.
   b. The equations in lines 109, 100 and 110 do not reflect Fig.2, and vice-versa. The scheme shown in Fig. 2 should also be reviewed with respect to connections.
   c. In line 157, when you say "RAP-Cell" do you mean "RAB"? Same in Figure 5, how does "RAM" use "RAP"? (Conflict with Figure 2 and the following equations).
5. At the beginning of section 3.3, part of the text is missing, perhaps comments on Figures 3a and 3b.
6. In section 4.1, please describe the dataset: data type, variable, instrument; rain or reflectivity; time period, etc.?
7. The equation 8 (page 10) is not a Z-R relation; it looks like a scale conversion. The Z-R relation is in the form of a power law $z = aR^b$.
8. In section 4.2, you should inform the range of the evaluation metrics.
9. In line 190, with "SST-LSTM" do you mean "RAP-Net"?
10. Are the results in Tables 1 to 4 calculated over the entire test set?
11. You should consider joining Table 1 with 2 and 3 with 4.
12. In line 195 "Besides, the proposed model has significant superiority especially for the nowcasting in heavy rainfall regions. Because…" you should join these sentences.
13. In line 199, what is the execution time of each compared model: PredRNN, PredRNN++ and RAP-Net?
14. In line 215, please give more arguments to state that your model is better in high thresholds.
15. In Figures 7 and 9, what is being shown, rain or reflectivity? Please fill in the caption.
16. In the conclusion (section 5), in future work, could you provide more details on how you intend to use more layers? What do you think about computational resources and execution time?

Other revisions:

1. The text is well written and comprehensible, but some sentences are short and break. The authors should join a few subsequent sentences together, using connectors and punctuation. Besides, there are some typological errors, such as: lines 52, 70, 93, 133, 148, 180, 195, Figure 3.
2. Separate citations from the text with parentheses. For example: line 16 (Shi et al., 2017); lines 22 and 23 (Shi et al., 2017; Ayzel et al., 2020; Li et al., 2021).
3. Use the acronyms RAB and RAM throughout the text in place of "Region Attention" (ex. line 218)
4. Review punctuation in lines 73, 74 and 75 "Our proposed method is different …"
5. Separate the terms in Figure 1 with commas, it looks like they are multiplying.

---

## Author Comment (AC1)

**Response to Reviewer #1**

**REVIEWER # 1**

*We must thank Reviewer #1 for providing us with useful comments for improving the quality of the paper. We have gone through the comments and have made revisions accordingly.*

Comments to the Author
Predictive Network. Through experiments, it can be confirmed that the model has some improvements in middle intensity rainfall prediction. However, there are still some problems need to be solved:

1.    The mechanism of RAB and RAM improving the prediction accuracy of middle intensity rainfall rather than other rainfall is not explained.
*Reply: Thanks for your question. From tables 3, we can find that the HSS and CSI of those models with RAB and RAM ($RAP - Cell_x$, $RAP - Cell_h$, RAP-Cell) are higher than the PredRNN model in the lowest threshold 5dBZ, which shows that RAB and RAM also can improve the accuracy in other rainfall.*

2.    The $\hat{X}_2$, $\hat{X}_{T-1}$, $\hat{X}_T$ in Figure 1 is not explained.
*Reply: Thanks for pointing it. We have added explanations about $\hat{X}_2$, $\hat{X}_{T-1}$, $\hat{X}_T$ as follows:*

*"At any timestamp t, model predicts a radar map $\hat{X}_{t+1}$ at the next timestamp t+1 according to the current radar map $X_t$ and historical radar sequence $X_{0:t}$."*

3.    The authors mention the improvement of RAM parameter performance, but lack discussion of RAB performance. In RAB, the global information is strengthened by self-attention mechanism, which brings more computation and parameters, and improves the prediction accuracy. Compared with the increase of computation, does this increase in accuracy meet expectation?
*Reply: Thanks for your question. In the subsection of ablation study, we discuss both of RAB and RAM as follows:*

*"To investigate the influence of various modules, we conduct an ablation study to discuss the effectiveness of Region Attention Block to the current input and the last hidden state. The result of evaluations is shown in Tables 3 and 4. $RAP - Cell_x$ and $RAP - Cell_h$ denote the PredRNN model embedding the RAB into the input and hidden state, respectively. RAP-Cell model is the combination of $RAP - Cell_x$ and $RAP - Cell_h$, and can also be regarded as RAP-Net without RAM. The results of $RAP - Cell_x$ and $RAP - Cell_h$, are higher than PredRNN, which shows the advantage of introducing Region Attention Block.  Specially, the $RAP - Cell_h$, significantly reduces the error according to MAE. Besides, the HSS, CSI and SSIM of RAP-Cell have significant improvements particularly when threshold $\tau$ is 40dBZ, which*

*implies that RAB simultaneously employed in the input and hidden state contributes to the prediction in the heavy rainfall regions. Moreover, by comparing the RAP-Cell and RAP-Net, we find that the RAM can enhance the accuracy of nowcasting especially in the area with middle-intensity rainfall.*

*Similarly, we also plot Figure 9 to show experimental results of all models against different nowcast lead times. We can see that RAP-Net delivers more promising result when the threshold increases, which demonstrates the effectiveness of combining RAB and RAM in terms of long-term prediction in high reflectivity area. The performance of RAM can be shown by comparing RAP-Cell and RAP-Net. We notice that the introduction of RAM can improve the prediction in the region of middle rainfall intensity. Besides, $RAP-Cell_x$ and $RAP-Cell_h$ embed RAB in the current input and the hidden state, respectively. Their performance is better than the original model PredRNN, especially in 20dBZ threshold. It shows the superiority of RAB. "*

*In the application of precipitation nowcasting, the predictions need to be generated within six minutes. Because the observed radar echo map is generated every six minutes. Therefore, the computation only satisfies a requirement that the time cost should be lower than six minutes. As for the proposed model, the generation of predictions only cost 1s. The increase of computation does not cause influence the precipitation nowcasting.*

4.    In Figure 3, what are the specific advantages of regional attention similarity matrix over other attention?

*Reply: Thanks for your question. Figure3 shows three types of attention methods. Traditional attention similarity in Figure3 (a) compares the difference between pixels. The attention similarity from Vision Transformer in Figure3 (b) compares the difference between regions with fixed size and position. The attention similarity from Region Attention (ours) in Figures (c) compares the difference between regions with flexible size and position. Considering that the shape reflectivity of radar echo is irregular and distributed in different places, attention manner in our method can capture the correlation between the different radar echoes better. Therefore, proposed region attention has a better spatiotemporal ability. Besides, we have directly shown specific advantages of regional attention similarity matrix in our manuscript as follows:*

*"Traditional attention mechanism calculates in Figure 3 (a) the similarity between different pixels and the attention manner in from Vision Transformer in Figure 3 (b) compares different regions in fixed location. Different from both, the attention similarity from Region Attention (ours) in Figure 3 (c) compares the difference between regions with flexible size and position. Due to the irregular shape of radar echo and different distribution, RAB can capture the correlation between the different radar echoes better. Therefore, the introduction of this block can improve the spatiotemporal ability of model.*

5.    Page 7 Line 128, "$\cdots Q_s \in R^{B \times N \times c*h*w}$, $K_s \in R^{B \times N \times c*h*w}$ and $V_s \in R^{B \times N \times c*h*w}$ are

fed…". What does tensor range mean?

*Reply: Thanks for your question. The types of the above tensors ($Q_s, K_s, V_s$) all belong to real numbers (R), Their size of these tensors all are [ $B \times N \times (c * h * w)$ ] with three dimensions.*

6. Please give the specific parameters of RAB and RAM.

*Reply: Thanks for your advice, we have offered the specific parameter of RAB and RAM as follows:*

*"It utilizes four layers RAP-Units as shown in Figure 1 and the parameters setting of each RAP-Unit are shown in Table 1."*

**Table 1.** The parameters setting of RAP-Unit

| Attention Type | Name | Kernel | Stride | Pad | Ch I/O | In Res | Out Res | Type |
|---|---|---|---|---|---|---|---|---|
| | $CNN_c$ | $5 \times 5$ | $1 \times 1$ | $2 \times 2$ | 64/64 | $32 \times 32$ | $32 \times 32$ | Conv |
| | $CNN_{qk}$ | $4 \times 4$ | $4 \times 4$ | $0 \times 0$ | 64/8 | $8 \times 8$ | $8 \times 8$ | Conv |
| Region Attention Block | $CNN_v$ | $5 \times 5$ | $1 \times 1$ | $2 \times 2$ | 64/64 | $32 \times 32$ | $32 \times 32$ | Conv |
| | $Lin_q$ | - | - | - | - | 512 | 512 | Linear |
| | $Lin_k$ | - | - | - | - | 512 | 512 | Linear |
| Recall Attention Mechanism | CNN | $5 \times 5$ | $1 \times 1$ | $2 \times 2$ | 14/64 | $32 \times 32$ | $32 \times 32$ | Conv |
| | $CNN_x$ | $5 \times 5$ | $1 \times 1$ | $2 \times 2$ | 64/448 | $32 \times 32$ | $32 \times 32$ | Conv |
| | $CNN_h$ | $5 \times 5$ | $1 \times 1$ | $2 \times 2$ | 64/256 | $32 \times 32$ | $32 \times 32$ | Conv |
| RNN unit | $CNN_m$ | $5 \times 5$ | $1 \times 1$ | $2 \times 2$ | 64/192 | $32 \times 32$ | $32 \times 32$ | Conv |
| | $CNN_o$ | $5 \times 5$ | $1 \times 1$ | $2 \times 2$ | 64/128 | $32 \times 32$ | $32 \times 32$ | Conv |
| | $CNN_{last}$ | $1 \times 1$ | $1 \times 1$ | $0 \times 0$ | 128/64 | $32 \times 32$ | $32 \times 32$ | Conv |

7. Page 8 Line 159. The authors regard $X_h^1$ as $K_c$ and $V_c$, then regard $H'_t$ as $Q_c$. Please explain the reason for this.

*Reply: Thanks for your question. The original output of RAP-Cell ($H'_t$) does not include long-term spatiotemporal representation because its update only utilizes the feature maps at the last time. To extract long-term spatiotemporal representation, we propose Recall Attention Mechanism (RAM) to retrieve the information of $X_h^l$ which convoluted from all inputs sequences $X_h^{l-1}$. From the equation (7), we can see $V_c$ can be extracted according to the $f(Q_c, K_c^T)$, where $Q_c$ decide how to explore the $V_c$ by dot-producing with $K_c$. Therefore, by RAM, the original output of RAP-Cell ($H'_t$) can capture long-term spatiotemporal information. To explain the reason, we have added some explanation in our manuscript as follows:*

*"From Eq. 7, we can see that the $V_c$ can be extracted according to the $f(Q_c, K_c^T)$, where $Q_c$ decides how to explore the $V_c$ by dot-producing with $K_c$. Therefore, in $l_{th}$ layer, the*

original output $H'^l_t$ of RAP-Cell can be regarded as query $Q_c$ to explore long-term spatiotemporal representations $X^l_h$ that is key $K_c$ and value $V_c$."

8.    Page 8 Line 163. "…the new output $H_t$ has recalled all original historical representation and long-term dependencies can be effectively preserved. Besides, the size of the long-memory feature map $X_h$ is fixed at any time." The reason why self-attention can achieve this effect is not explained.

*Reply: Thanks for your question. We have added some content to explain why $H_t$ can recall all original historical representation as follows:*

*"From Eq. 7, we can see that the $V_c$ can be extracted according to the $f(Q_c, K^T_c)$, where $Q_c$ decides how to explore the $V_c$ by dot-producing with $K_c$. Therefore, in $l_{th}$ layer, the original output $H'^l_t$ of RAP-Cell can be regarded as query $Q_c$ to explore long-term spatiotemporal representations $X^l_h$ that is key $K_c$ and value $V_c$."*

*Besides, the fixed size of the long-memory feature map $X_h$ at any time has no relationship with self-attention. Because the size of $X_h$ is predefined and corresponding content at different timestamps are fed into $X_h$. We have added the explanation as follows:*

*"Because the size of $X_h$ is predefined and corresponding content at different timestamps are fed into $X_h$"*

9.    As for RAM having more advantages than the recall mechanism of EIDETIC 3D LSTM, the paper lacks important experiment to prove it.

*Reply: Thanks for your advice. We have added an experiment about EIDETIC 3D LSTM applied in precipitation nowcasting in Tables 2 as follows::*

**Table 2.** Comparison results on RadarCIKM in terms of HSS, CSI, SSIM, and MAE

| Methods | HSS ↑ | | | | CSI ↑ | | | | MAE ↓ | SSIM ↑ |
|---|---|---|---|---|---|---|---|---|---|---|
| | 5dBZ | 20dBZ | 40dBZ | avg | 5dBZ | 20dBZ | 40dBZ | avg | | |
| ConvLSTM Xingjian et al. (2015) | 0.7031 | 0.4857 | 0.1470 | 0.4453 | 0.7663 | 0.4092 | 0.0801 | 0.4186 | 5.97 | 0.6334 |
| ConvGRU Shi et al. (2017) | 0.6816 | 0.4827 | 0.1225 | 0.4289 | 0.7522 | 0.3952 | 0.0657 | 0.4043 | 6.00 | 0.6338 |
| TrajGRU Shi et al. (2017) | 0.6809 | 0.4945 | 0.1907 | 0.4553 | 0.7466 | 0.4028 | 0.1061 | 0.4185 | 5.90 | 0.6424 |
| DFN Jia et al. (2016) | 0.6772 | 0.4719 | 0.1306 | 0.4266 | 0.7489 | 0.3771 | 0.0704 | 0.3988 | 6.03 | 0.6268 |
| PredRNN Wang et al. (2017) | 0.7082 | 0.4915 | 0.1639 | 0.4606 | 0.7692 | 0.4051 | 0.0901 | 0.4215 | 5.42 | 0.6887 |
| PredRNN++ Wang et al. (2018a) | 0.7061 | 0.5047 | 0.1710 | 0.4548 | 0.7642 | 0.4176 | 0.0940 | 0.4253 | 5.44 | 0.6851 |
| E3D-LSTM Wang et al. (2018b) | 0.7111 | 0.4810 | 0.1361 | 0.4427 | 0.7720 | 0.4060 | 0.0734 | 0.4171 | 5.51 | 0.6958 |
| MIM Wang et al. (2019) | 0.7052 | 0.5166 | 0.1858 | 0.4692 | 0.7628 | 0.4279 | 0.1034 | 0.4313 | 5.47 | 0.6796 |
| PhyDNet Guen and Thome (2020) | 0.6741 | 0.4709 | 0.1832 | 0.4427 | 0.7402 | 0.4003 | 0.1017 | 0.4141 | 6.25 | 0.6443 |
| SA-ConvLSTM Lin et al. (2020) | **0.7118** | 0.4861 | 0.1582 | 0.4520 | **0.7725** | 0.4161 | 0.0870 | 0.4252 | 5.71 | 0.6709 |
| PFST-LSTM Luo et al. (2020) | 0.7045 | 0.5071 | 0.2218 | 0.4778 | 0.7680 | 0.4175 | 0.1257 | 0.4371 | 5.82 | 0.6367 |
| CMS-LSTM Chai et al. (2021) | 0.6835 | 0.4605 | 0.1720 | 0.4387 | 0.7567 | 0.3788 | 0.0948 | 0.4101 | 5.95 | 0.6496 |
| RAP-Net | 0.7117 | **0.5116** | **0.2293** | **0.4842** | 0.7666 | **0.4305** | **0.1307** | **0.4426** | **5.37** | **0.7019** |

*Besides, we have added the result of E3D-LSTM in Figure 6 as follows:*

[Figure]

(a) HSS $\tau$=5;  (b) CSI $\tau$=5;

(c) HSS $\tau$=20;  (d) CSI $\tau$=20;

(e) HSS $\tau$=40;  (f) CSI $\tau$=40;

**Figure 6.** The HSS and CSI scores of different nowcase lead time values. (Best view in color)

*Moreover, the new visualization results are shown in Figure 7 as follows:*

[Figure]

ConvLSTM

ConvGRU

TrajGRU

DFN

PredRNN

PredRNN++

E3D-LSTM

MIM

PhyDNet

SA-ConvLSTM

PFST-LSTM

RAP-Net

CMS-LSTM

-5-0  0-5  5-10  10-15 15-20 20-25 25-30 30-35 35-40 40-45 45-50 50-55 55-60 60-65

**Figure 7.** The first row is the ground truth and reminders are the predictions of various methods on an example from the RadarCIKM dataset (Best view in color)

10. What is the resolution of the image processed before experiments?

*Reply: Thanks for your question. The resolution of image is 101 x 101. We have added this description as follows:*

*"Each sequence contains 15 continual observations within 90 minutes, where the spatial and temporal resolution of each map is 101 x 101 and six minutes, respectively."*

Minor comments:

1. Page 3 Line 85, "It utilizes the structure of PredRNN …". When the abstract and previous part have been around ConvRNN method, PredRNN is mentioned here. Is it possible to explain the relationship between them?

*Reply: Thanks for your question. The ConvRNN is the general name for a series of algorithms that combines convolution and recurrent neural networks. Here PredRNN is a classical method in ConvRNN.*

2. Page 11 Line 202, "… which implies the Region Attention can improve …". Is it possible to write RAB and RAM together, not just Region Attention?

*Reply: Thanks for your advice. We have modified this sentence as follows:*

*"Nevertheless its performance is poor in the highest threshold (40dBZ), which implies the RAB and RAM can improve the prediction in the area with high radar echo compared to traditional attention mechanism."*

---

## Author Comment (AC2)

**Response to Reviewer #2**

**REVIEWER # 2**

*We must thank Reviewer #2 for providing us with useful comments to improve this article. We have gone through the comments and made revisions accordingly.*

Comments to the Author

The paper describes a RAP-Net network that can be used for radar echo extrapolation. Experiments demonstrate the effectiveness of this method. The authors are suggested to supplement the experimental comparison of high-intensity echoes.

*Reply: Thanks for your advice. The performance of RAP-Net in high-intensity echoes can be represented from three aspects. Firstly, from the evaluation metrics in Table 2, the HSS and CSI of RAP-Net in the highest thresholds (40dBZ) are higher than other models. Secondly, from Figures 6 (e) and (f), the predictions of RAP-Net keep the best HSS and CSI in 40 dBZ under most of the lead time. Especially, in the last prediction, the HSS and CSI of RAP-Net are obviously higher than other models. Finally, from Figure 7, we can see that only the RAP-Net model predicts color regions. It implies that the proposed model predicts better than other models in heavy rainfall areas. The above three observations jointly confirm that our model is better in high thresholds. The table 2, Figure 6, and Figure 7 can be respectively shown as follows:*

**Table 2.** Comparison results on RadarCIKM in terms of HSS, CSI, SSIM, and MAE

| Methods | HSS ↑ | | | | CSI ↑ | | | | MAE ↓ | SSIM ↑ |
|---|---|---|---|---|---|---|---|---|---|---|
| | 5dBZ | 20dBZ | 40dBZ | avg | 5dBZ | 20dBZ | 40dBZ | avg | | |
| ConvLSTM Xingjian et al. (2015) | 0.7031 | 0.4857 | 0.1470 | 0.4453 | 0.7663 | 0.4092 | 0.0801 | 0.4186 | 5.97 | 0.6334 |
| ConvGRU Shi et al. (2017) | 0.6816 | 0.4827 | 0.1225 | 0.4289 | 0.7522 | 0.3952 | 0.0657 | 0.4043 | 6.00 | 0.6338 |
| TrajGRU Shi et al. (2017) | 0.6809 | 0.4945 | 0.1907 | 0.4553 | 0.7466 | 0.4028 | 0.1061 | 0.4185 | 5.90 | 0.6424 |
| DFN Jia et al. (2016) | 0.6772 | 0.4719 | 0.1306 | 0.4266 | 0.7489 | 0.3771 | 0.0704 | 0.3988 | 6.03 | 0.6268 |
| PredRNN Wang et al. (2017) | 0.7082 | 0.4915 | 0.1639 | 0.4606 | 0.7692 | 0.4051 | 0.0901 | 0.4215 | 5.42 | 0.6887 |
| PredRNN++ Wang et al. (2018a) | 0.7061 | 0.5047 | 0.1710 | 0.4548 | 0.7642 | 0.4176 | 0.0940 | 0.4253 | 5.44 | 0.6851 |
| E3D-LSTM Wang et al. (2018b) | 0.7111 | 0.4810 | 0.1361 | 0.4427 | 0.7720 | 0.4060 | 0.0734 | 0.4171 | 5.51 | 0.6958 |
| MIM Wang et al. (2019) | 0.7052 | 0.5166 | 0.1858 | 0.4692 | 0.7628 | 0.4279 | 0.1034 | 0.4313 | 5.47 | 0.6796 |
| PhyDNet Guen and Thome (2020) | 0.6741 | 0.4709 | 0.1832 | 0.4427 | 0.7402 | 0.4003 | 0.1017 | 0.4141 | 6.25 | 0.6443 |
| SA-ConvLSTM Lin et al. (2020) | **0.7118** | 0.4861 | 0.1582 | 0.4520 | **0.7725** | 0.4161 | 0.0870 | 0.4252 | 5.71 | 0.6709 |
| PFST-LSTM Luo et al. (2020) | 0.7045 | 0.5071 | 0.2218 | 0.4778 | 0.7680 | 0.4175 | 0.1257 | 0.4371 | 5.82 | 0.6367 |
| CMS-LSTM Chai et al. (2021) | 0.6835 | 0.4605 | 0.1720 | 0.4387 | 0.7567 | 0.3788 | 0.0948 | 0.4101 | 5.95 | 0.6496 |
| RAP-Net | 0.7117 | **0.5116** | **0.2293** | **0.4842** | 0.7666 | **0.4305** | **0.1307** | **0.4426** | **5.37** | **0.7019** |

(a) HSS $\tau=5$;  (b) CSI $\tau=5$;

(c) HSS $\tau=20$;  (d) CSI $\tau=20$;

(e) HSS $\tau=40$;  (f) CSI $\tau=40$;

**Figure 6.** The HSS and CSI scores of different nowcase lead time values. (Best view in color)

12

[Figure]

**Figure 7.** The first row is the reflectivity of ground truth and reminders are the predicted reflectivity of various methods on an example from the RadarCIKM dataset (Best view in color)

---

## Author Comment (AC3)

**Response to Reviewer #3**

**REVIEWER #3**

We must thank Reviewer #3 for providing us with useful comments to improve this article. We have gone through the comments and made revisions accordingly.

**Comments to the Author**

To predict weather radar image sequences, the authors propose the model RAP-Net that add attention modules (RAB and RAM) in a ConvRNN model, in order to improve forecasting in the area with heavy rainfall (RAB) and improve the long-term spatiotemporal representation ability (RAM).

 In general, rainfall intensity is classified as light, moderate and heavy (AMS glossary "Rain": https://glossary.ametsoc.org/wiki/Rain), not as "strong" (line 7) or "middle" (lines 10, 44, 53 and 243). Please fix this throughout the text.

Reply: Thanks for your advice. We have fixed in our manuscript.

2. In line 22, what are the limitations of the traditional approach you are referring to? *Reply: Thanks for your question. As the previous sentence says, the limitation of traditional methods is that they do not exploit abundant historical observations as follows:*

"However, these methods do not exploit abundant historical observations."

3. In line 39, the sentence "where the similar semantics gathered in the same tensor." sounds incomplete.

Reply: Thanks for pointing it. We have rewritten it as follows:

"RAB classifies each feature map into equal-sized tensors and each tensor gatherers a similar semantic."

4. I suggest reviewing all equations to simplify the notation and unify the letters of all equations (sections 3.2, 3.3 and 3.4):

a. The notation adopted by the terms of equations is confusing (sections 3.2, 3.3 and 3.4):

1. In line 89, what are the letters "I" and "h" in  $X_h^l$ ?

2. Is  $X_h^l = X_t$ ?

3. Etc.

*Reply: Thanks for your question. Here, the "I" indicates the number of layers. In the bottom layer,*  $X_h^l$  *is the result after convolution by historical input sequences*  $X_{0:t}$ *. In the other layers, the*  $X_h^l$  *is the result after convolution by the*  $X_h^{l-1}$ *. Therefore, the*  $X_h^l$  *is not*  $X_t$ *. We have added some explanations in our manuscript as follows:*

"Similarly, in the  $l_{th}$  layer, the input of the long-memory hidden state is the  $X_h^l$ . In the bottom layer,  $X_h^l$  is the result after convolution by historical input sequences  $X_{0:t}$ . In the other layers, the  $X_h^l$  is the result after convolution by the  $X_h^{l-1}$ . By RAM, the long-term historical representation can be delivered to the next layer."

b. The equations in lines 109, 100 and 110 do not reflect Fig.2, and vice-versa. The scheme shown in Fig. 2 should also be reviewed with respect to connections.

Reply: Thanks for pointing it. We have corrected errors in equation 2 and Fig.2 as follows:

```
\begin{split} X_t'^l &= RAB(X_t^l), \\ H_{t-1}'^l &= RAB(H_{t-1}^l), \\ i_t &= \sigma(W_{xi} * X_t'^l + W_{hi} * H_{t-1}'^l + b_i), \\ g_t &= tanh(W_{xg} * X_t'^l + W_{hg} * H_{t-1}'^l + b_g), \\ f_t &= \sigma(W_{xf} * X_t'^l + W_{hf} * H_{t-1}'^l + b_f), \\ i_t' &= \sigma(W_{xi}' * X_t'^l + W_{mi} * M_t^{l-1} + b_i'), \\ g_t' &= tanh(W_{xg}' * X_t'^l + W_{mg} * M_t^{l-1} + b_g'), \\ f_t' &= \sigma(W_{xf}' * X_t'^l + W_{mf} * M_t^{l-1} + b_f'), \\ C_t^l &= i_t \circ g_t + f_t \circ C_{t-1}^l, \\ M_t^l &= i_t' \circ g_t' + f_t' \circ M_t^{l-1}, \\ o_t &= \sigma(W_{xo} * X_t'^l + W_{ho} * H_{t-1}' + W_{co} * C_t^l + W_{mo} * M_t^l + b_o), \\ H_t^l &= o_t \circ tanh(W_{1 \times 1} * [X_t'^l, M_t^k]), \\ H_t^l, X_h^l &= RAM(H_t^l, X_h^{l-1} * W_l), \end{split}
```

Figure 2. The internal structure of the Region Attention Predictive Unit (RAP-Unit)

c. In line 157, when you say "RAP-Cell" do you mean "RAB"? Same in Figure 5, how does "RAM" use "RAP"? (Conflict with Figure 2 and the following equations).

Reply: Thanks for your question. In this paper, we propose two submodules. They are RAB and RAM. By embedding these two submodules, the RAP unit is built as Figure 2 shown. In other words, the RAP unit contains the RAM and RAB. The new equation (2) reflects the calculating process of the RAP unit as Figure 2 shown. The new equation (2) and Figure 2 are

**shown as follows:**

```
\begin{split} X_t^{\prime l} &= RAB(X_t^l), \\ H_{t-1}^{\prime l} &= RAB(H_{t-1}^l), \\ i_t &= \sigma(W_{xi} * X_t^{\prime l} + W_{hi} * H_{t-1}^{\prime l} + b_i), \\ g_t &= tanh(W_{xg} * X_t^{\prime l} + W_{hg} * H_{t-1}^{\prime l} + b_g), \\ f_t &= \sigma(W_{xf} * X_t^{\prime l} + W_{hf} * H_{t-1}^{\prime l} + b_f), \\ i_t' &= \sigma(W_{xi} * X_t^{\prime l} + W_{mi} * M_t^{l-1} + b_i'), \\ g_t' &= tanh(W_{xg} * X_t^{\prime l} + W_{mg} * M_t^{l-1} + b_g'), \\ f_t' &= \sigma(W_{xf} * X_t^{\prime l} + W_{mf} * M_t^{l-1} + b_f'), \\ C_t^l &= i_t \circ g_t + f_t \circ C_{t-1}^l, \\ M_t^l &= i_t' \circ g_t' + f_t' \circ M_t^{l-1}, \\ o_t &= \sigma(W_{xo} * X_t^{\prime l} + W_{ho} * H_{t-1}^{\prime l} + W_{co} * C_t^l + W_{mo} * M_t^l + b_o), \\ H_t^l &= o_t \circ tanh(W_{1 \times 1} * [X_t^{\prime l}, M_t^k]), \\ H_t^l, X_h^l &= RAM(H_t^l, X_h^{l-1} * W_l), \end{split}
```

(2)

Besides, RAP-Cell is the RAP unit without the RAM sub-module. Here, we use Figure 5 to explain how to introduce RAM into the proposed predicted unit. To clear it, we give the explanation of RAP-Cell in the caption of Figure 5 as follows:

---

## Author Response (AR1)

**Reply Letter**

**Response to Reviewer #1**

**REVIEWER # 1**

We must thank Reviewer #1 for providing us with useful comments for improving the quality of the paper. We have gone through the comments and have made revisions accordingly.

Comments to the Author

Predictive Network. Through experiments, it can be confirmed that the model has some improvements in middle intensity rainfall prediction. However, there are still some problems need to be solved:

1. The mechanism of RAB and RAM improving the prediction accuracy of middle intensity rainfall rather than other rainfall is not explained.

*Reply:* Thanks for your question. From tables 3, we can find that the HSS and CSI of those models with RAB and RAM ( $RAP - Cell_x$ ,  $RAP - Cell_h$ , RAP - Cell) are higher than the PredRNN model in the lowest threshold 5dBZ, which shows that RAB and RAM also can improve the accuracy in other rainfall.

2. The  $X_2$ ,  $X_{T-1}$ ,  $X_T$  in Figure 1 is not explained.

Reply: Thanks for pointing it. We have added explanations about  $X_2$ ,  $X_{T-1}$ ,  $X_T$  as follows:

"At any timestamp t, model predicts a radar map  $X_{t+1}$  at the next timestamp t+1 according to the current radar map  $X_t$  and historical radar sequence  $X_{0:t}$ ."

The details can be found in line 86 on page 3.

3. The authors mention the improvement of RAM parameter performance, but lack discussion of RAB performance. In RAB, the global information is strengthened by self-attention mechanism, which brings more computation and parameters, and improves the prediction accuracy. Compared with the increase of computation, does this increase in accuracy meet expectation? *Reply: Thanks your question. In the subsection of ablation study, we discuss both of RAB and RAM as follows:*

"To investigate the influence of various modules, we conduct an ablation study to discuss the effectiveness of Region Attention Block to the current input and the last hidden state. The result

of evaluations is shown in Tables 3 and 4.  $RAP - Cell_x$  and  $RAP - Cell_h$  denote the PredRNN model embedding the RAB into the input and hidden state, respectively. RAP-Cell model is the combination of  $RAP - Cell_x$  and  $RAP - Cell_h$ , and can also be regarded as RAP-Net without RAM.

The results of  $RAP - Cell_x$  and  $RAP - Cell_h$ , are higher than PredRNN, which shows the

advantage of introducing Region Attention Block. Specially, the  $RAP - Cell_h$ , significantly reduces the error according to MAE. Besides, the HSS, CSI and SSIM of RAP-Cell have significant improvements particularly when threshold  $\tau$  is 40dBZ, which implies that RAB simultaneously employed in the input and hidden state contributes to the prediction in the heavy rainfall regions. Moreover, by comparing the RAP-Cell and RAP-Net, we find that the RAM can enhance the accuracy of nowcasting especially in the area with middle-intensity rainfall.

Similarly, we also plot Figure 9 to show experimental results of all models against different nowcast lead times. We can see that RAP-Net delivers more promising result when the threshold increases, which demonstrates the effectiveness of combining RAB and RAM in terms of long-term prediction in high reflectivity area. The performance of RAM can be shown by comparing RAP-Cell and RAP-Net. We notice that the introduction of RAM can improve the prediction in the region of

middle rainfall intensity. Besides,  $RAP - Cell_x$  and  $RAP - Cell_h$  embed RAB in the current input and the hidden state, respectively. Their performance is better than the original model PredRNN, especially in 20dBZ threshold. It shows the superiority of RAB. "

The details can be found in line 233 on page 14...

In the application of precipitation nowcasting, the predictions need to be generated within six minutes. Because the observed radar echo map is generated every six minutes. Therefore, the computation only satisfies a requirement that the time cost should be lower than six minutes. As for the proposed model, the generation of predictions only cost 1s. The increase of computation does not cause influence the precipitation nowcasting.

4. In Figure 3, what are the specific advantages of regional attention similarity matrix over other attention?

Reply: Thanks for your question. Figure3 shows three types of attention methods. Traditional attention similarity in Figure3 (a) compares the difference between pixels. The attention similarity from Vision Transformer in Figure3 (b) compares the difference between regions with fixed size and position. The attention similarity from Region Attention (ours) in Figures (c) compares the difference between regions with flexible size and position. Considering that the shape reflectivity of radar echo is irregular and distributed in different places, attention manner in our method can capture the correlation between the different radar echoes better. Therefore, proposed region attention has a better spatiotemporal ability. Besides, we have directly shown specific advantages of regional attention similarity matrix in our manuscript as follows:

"Traditional attention mechanism calculates in Figure 3 (a) the similarity between different pixels and the attention manner in from Vision Transformer in Figure 3 (b) compares different regions in fixed location. Different from both, the attention similarity from Region Attention (ours) in Figure 3 (c) compares the difference between regions with flexible size and position. Due to the irregular shape of radar echo and different distribution, RAB can capture the correlation between the different radar echoes better. Therefore, the introduction of this block can improve the spatiotemporal ability of model.

The details can be found in line 148 on page 8.

5. Page 7 Line 128, " $\cdots Q_s \in R^{B \times N \times c^*h^*w}$ ,  $K_s \in R^{B \times N \times c^*h^*w}$  and  $V_s \in R^{B \times N \times c^*h^*w}$  are fed...". What does tensor range mean?

Reply: Thanks for your question. The types of the above tensors  $(Q_s, K_s, V_s)$  all belong to real numbers (R), Their size of these tensors all are  $[B \times N \times (c * h * w)]$  with three dimensions.

6. Please give the specific parameters of RAB and RAM.

Reply: Thanks for your advice, we have offered the specific parameter of RAB and RAM as follows:

"It utilizes four layers RAP-Units as shown in Figure 1 and the parameters setting of each RAP-Unit are shown in Table 1."

| Attention Type             | Name                       | Kernel       | Stride       | Pad          | Ch I/O | In Res         | Out Res        | Туре   |
|----------------------------|----------------------------|--------------|--------------|--------------|--------|----------------|----------------|--------|
| Region Attention Block     | $CNN_c$                    | $5 \times 5$ | $1 \times 1$ | $2 \times 2$ | 64/64  | 32 × 32        | 32 × 32        | Conv   |
|                            | $\mathrm{CNN}_{qk}$        | $4 \times 4$ | $4 \times 4$ | 0 	imes 0    | 64/8   | $8 \times 8$   | $8 \times 8$   | Conv   |
|                            | $\mathrm{CNN}_v$           | $5 \times 5$ | $1 \times 1$ | $2 \times 2$ | 64/64  | $32 \times 32$ | $32 \times 32$ | Conv   |
|                            | $\operatorname{Lin}_q$     | -            | -            | -            | -      | 512            | 512            | Linear |
|                            | $Lin_k$                    | -            | -            | -            | -      | 512            | 512            | Linear |
| Recall Attention Mechanism | CNN                        | $5 \times 5$ | $1 \times 1$ | $2 \times 2$ | 14/64  | $32 \times 32$ | $32 \times 32$ | Conv   |
| RNN unit                   | $\text{CNN}_x$             | $5 \times 5$ | $1 \times 1$ | $2 \times 2$ | 64/448 | $32 \times 32$ | $32 \times 32$ | Conv   |
|                            | $\mathrm{CNN}_h$           | $5 \times 5$ | $1 \times 1$ | $2 \times 2$ | 64/256 | $32 \times 32$ | $32 \times 32$ | Conv   |
|                            | $\text{CNN}_m$             | $5 \times 5$ | $1 \times 1$ | $2 \times 2$ | 64/192 | $32 \times 32$ | $32 \times 32$ | Conv   |
|                            | $CNN_o$                    | $5 \times 5$ | $1 \times 1$ | $2 \times 2$ | 64/128 | $32 \times 32$ | $32 \times 32$ | Conv   |
|                            | CNN last | $1 \times 1$ | $1 \times 1$ | 0 	imes 0    | 128/64 | $32 \times 32$ | $32 \times 32$ | Conv   |

Table 1. The parameters setting of RAP-Unit

The details can be found on page 10.

7. Page 8 Line 159. The authors regard  $X_h^7$  as  $K_c$  and  $V_c$ , then regard  $H'_t$  as  $Q_c$ . Please explain the reason for this.

Reply: Thanks for your question. The original output of RAP–Cell ( $H'_t$ ) does not include long–term spatiotemporal representation because its update only utilizes the feature maps at the last time. To extract long–term spatiotemporal representation, we propose Recall Attention Mechanism (RAM) to retrieve the information of  $X_h^l$  which convoluted from all inputs sequences  $X_h^{l-1}$ . From the equation (7), we can see  $V_c$  can be extracted according to the  $f(Q_{\sigma} K_c^T)$ , where  $Q_c$  decide how to explore the  $V_c$  by dot–producing with  $K_c$ . Therefore, by RAM, the original output of RAP–Cell ( $H'_t$ ) can capture long–term spatiotemporal information. To explain the reason, we have added some explanation in our manuscript as follows:

"From Eq. 7, we can see that the  $V_c$  can be extracted according to the  $f(Q_c, K_c^T)$ , where  $Q_c$  decides how to explore the  $V_c$  by dot-producing with  $K_c$ . Therefore, in  $I_{th}$  layer, the original output  $H_t^{\prime}$  of RAP-CeII can be regarded as query  $Q_c$  to explore long-term spatiotemporal representations  $X_h^{\prime}$  that is key  $K_c$  and value  $V_c$ ."

**The details can be found in line 168 on page 8.**

8. Page 8 Line 163. "...the new output  $H_t$  has recalled all original historical representation and long-term dependencies can be effectively preserved. Besides, the size of the long-memory feature map  $X_h$  is fixed at any time." The reason why self-attention can achieve this effect is not explained.

Reply: Thanks for your question. We have added some content to explain why  $H_t$  can recall all original historical representation as follows:

"From Eq. 7, we can see that the  $V_c$  can be extracted according to the  $f(Q_c, K_c^T)$ , where  $Q_c$  decides how to explore the  $V_c$  by dot-producing with  $K_c$ . Therefore, in  $I_{th}$  layer, the original

output  $H'_t$  of RAP-Cell can be regarded as query  $Q_c$ to explore long-term spatiotemporal representations  $X'_h$  that is key  $K_c$  and value  $V_c$ ."

The details can be found in line 168 on page 8.

Besides, the fixed size of the long-memory feature map  $X_h$  at any time has no relationship with self-attention. Because the size of  $X_h$  is predefined and corresponding content at different timestamps are fed into  $X_h$ . We have added the explanation as follows:

"Because the size of  $X_h$  is predefined and corresponding content at different timestamps are fed

into  $X_h$ "

The details can be found in line 172 on page 8.

9. As for RAM having more advantages than the recall mechanism of EIDETIC 3D LSTM, the paper lacks important experiment to prove it.

Reply: Thanks for your advice. We have added an experiment about EIDETIC 3D LSTM applied in precipitation nowcasting in Tables 2 as follows::

Table 2. Comparison results on RadarCIKM in terms of HSS, CSI, SSIM, and MAE

| Methods                         | HSS ↑  |                      |               |               |               | CS            | MAE↓   | SSIM ↑        |             |               |
|---------------------------------|--------|----------------------|---------------|---------------|---------------|---------------|--------|---------------|-------------|---------------|
|                                 | 5dBZ   | 20dBZ                | 40dBZ         | avg           | 5dBZ          | 20dBZ         | 40dBZ  | avg           |             |               |
| ConvLSTM Xingjian et al. (2015) | 0.7031 | 0.4857               | 0.1470        | 0.4453        | 0.7663        | 0.4092        | 0.0801 | 0.41 86       | 5.97        | 0.6334        |
| ConvGRU Shi et al. (2017)       | 0.6816 | 0.4827               | 0.1225        | 0.4289        | 0.7522        | 0.3952        | 0.0657 | 0.4043        | 6.00        | 0.6338        |
| TrajGRU Shi et al. (2017)       | 0.6809 | 0.4945               | 0.1907        | 0.4553        | 0.7466        | 0.4028        | 0.1061 | 0.4185        | 5.90        | 0.6424        |
| DFN Jia et al. (2016)           | 0.6772 | 0.4719               | 0.1306        | 0.4266        | 0.7489        | 0.3771        | 0.0704 | 0.3988        | 6.03        | 0.6268        |
| PredRNN Wang et al. (2017)      | 0.7082 | 0.4915               | 0.1639        | 0.4606        | 0.7692        | 0.4051        | 0.0901 | 0.4215        | 5.42 | 0.6887        |
| PredRNN++ Wang et al. (2018a)   | 0.7061 | 0.5047               | 0.1710        | 0.4548        | 0.7642        | 0.4176        | 0.0940 | 0.4253        | 5.44        | 0.6851        |
| E3D-LSTM Wang et al. (2018b)    | 0.7111 | 0.4810               | 0.1361        | 0.4427        | 0.7720 | 0.4060        | 0.0734 | 0.4171        | 5.51        | 0.6958 |
| MIM Wang et al. (2019)          | 0.7052 | 0.5166               | 0.1858        | 0.4692        | 0.7628        | 0.4279        | 0.1034 | 0.4313        | 5.47        | 0.6796        |
| PhyDNet Guen and Thome (2020)   | 0.6741 | 0.4709               | 0.1832        | 0.4427        | 0.7402        | 0.4003        | 0.1017 | 0.4141        | 6.25        | 0.6443        |
| SA-ConvLSTM Lin et al. (2020)   | 0.7118 | 0.4861               | 0.1582        | 0.4520        | 0.7725        | 0.4161        | 0.0870 | 0.4252        | 5.71        | 0.6709        |
| PFST-LSTM Luo et al. (2020)     | 0.7045 | $\underline{0.5071}$ | 0.2218 | 0.4778 | 0.7680        | 0.4175 | 0.1257 | 0.4371 | 5.82        | 0.6367        |
| CMS-LSTM Chai et al. (2021)     | 0.6835 | 0.4605               | 0.1720        | 0.4387        | 0.7567        | 0.3788        | 0.0948 | 0.4101        | 5.95        | 0.6496        |
| RAP-Net                         | 0.7117 | 0.5116               | 0.2293        | 0.4842        | 0.7666        | 0.4305        | 0.1307 | 0.4426        | 5.37        | 0.7019        |

Besides, we have added the result of E3D-LSTM in Figure 6 as follows:

---

## Referee Report (RR1)

Review

The authors have revised the model structure, parameters and principles in this manuscript. Through RAB and RAM, global and local spatial features are captured simultaneously, while spatiotemporal expressivity is enhanced. These two new mechanisms effectively improve the accuracy of rainfall prediction. However, there are still few problems:

1.Page2 Line50, '…the same echo with large-scale size and long-range movement between the adjacent time, more useful spatial information can be extracted, which leads to more accurate predictions in those regions…'. From the ground truth given in this paper, the non-heavy rainfall echoes also have a large-scale size and long-range movement between adjacent time. The reason why RAB has better effect on heavy rainfall needs to be explained

2. Page2 Line53, 'The representation of moderate and heavy rainfall intensity can be preserved in the predicted unit'. What is the reason that rainstorm information is easier to retain through RAM than others.

3. Page16 Table 3. The addition of RAM reduces the accuracy of 5dBZ. Please explain the reason.

---

## Author Response (AR3)

**Reply Letter**

**Response to Reviewer #1**

**REVIEWER # 1**

*We must thank Reviewer #1 for providing us with useful comments for improving the quality of the paper. We have gone through the comments and have made revisions accordingly.*

Comments to the Author

The authors have revised the model structure, parameters and principles in this manuscript. Through RAB and RAM, global and local spatial features are captured simultaneously, while spatiotemporal expressivity is enhanced. These two new mechanisms effectively improve the accuracy of rainfall prediction. However, there are still few problems:

1. Page2 Line50, '···the same echo with large-scale size and long-range movement between the adjacent time, more useful spatial information can be extracted, which leads to more accurate predictions in those regions···'. From the ground truth given in this paper, the non-heavy rainfall echoes also have a large-scale size and long-range movement between adjacent time. The reason why RAB has better effect on heavy rainfall needs to be explained

*Reply: Thanks for your question. As our paper mention:*

*"The attention similarity from Region Attention (ours) in Figure 3(c) compares the difference between regions with flexible size and position. Due to the irregular shape of radar echo and different distribution, RAB can capture the correlation between the different radar echoes better." (In line 150 on page 8.)*

*RAB has the advantage of extracting the spatial representation of the irregular-shaped object. In different ranges of reflectivity, we notice that radar echoes with high reflectivity have more stable appearance and shape. Therefore, their information is more easily extracted. We have explained it as follows:*

*"Especially, the information of radar echoes with high reflectivity is more easily extracted because they have a more stable appearance and shape. RAM has more contribution to improving the performance in these regions with heavy rainfall"*

*The details can be found in line 153 on page 8.*

2. Page2 Line53, 'The representation of moderate and heavy rainfall intensity can be preserved in the predicted unit'. What is the reason that rainstorm information is easier to retain through RAM than others.

*Reply: Thanks for your question. The function of RAM is to retain long-term spatial-temporal representations. These representations contain information about rainstorms even though their ratio is very low. The reason why RAM can retain these representations can be found in the structure of RAM as follows:*

[Figure]

*Specifically, there are two aspects to explain it. Firstly, in terms of the input of RAM, the original input of RAM is $X_{0:t}$. However, the input of traditional ConvRNN is $X_t$. In other words, the RAM extracts spatial-temporal representation from all historical radar echo maps instead of the current radar map. Besides, in terms of the internal structure of RAM, RAM applies a channel attention mechanism to recall all previous data. By using channel attention, all historical feature maps participate in the calculation and obtain the final output of RAM.*

3.  Page16 Table 3. The addition of RAM reduces the accuracy of 5dBZ. Please explain the reason.

*Reply: Thanks for your question. RAM module can preserve the long-term historical representations by attention mechanism. This mechanism can recall previous information by a similar metric. The more similar to the current representation, the higher probability to be*

*saved. However, we notice that some radar echoes with low reflectivity usually have more unstable changes compared with those echoes with moderate and high reflectivity. It leads to that the historical representations involving low reflectivity radar echo probably have more difference from the current representation. Therefore, it implies that their similarity is relatively low and the information of low reflectivity is hard to be preserved. The radar echoes with low reflectivity are hard to be predicted also can be shown from the visualization of predictions in Figure 9 (Comparing with RAP-Cell and RAP-Net). It is why the addition of RAM reduces the accuracy of 5 dBZ.*

[Figure]

**Figure 9.** The first row is the reflectivity of ground truth and reminders are the predicted reflectivity of different methods on an example from the RadarCIKM dataset (Best view in color)